# An integrated bioinformatics platform for investigating the human E3 ubiquitin ligase-substrate interaction network

Yang Li[1], Ping Xie[1,2], Liang Lu[1], Jian Wang[1], Lihong Diao[1,3], Zhongyang Liu[1], Feifei Guo[1], Yangzhige He[1], Yuan Liu[1], Qin Huang[3], Han Liang [4], Dong Li[1] & Fuchu He[1]

The ubiquitination mediated by ubiquitin activating enzyme (E1), ubiquitin conjugating enzyme (E2), and ubiquitin ligase (E3) cascade is crucial to protein degradation, transcription regulation, and cell signaling in eukaryotic cells. The high specificity of ubiquitination is regulated by the interaction between E3 ubiquitin ligases and their target substrates. Unfortunately, the landscape of human E3-substrate network has not been systematically uncovered. Therefore, there is an urgent need to develop a high-throughput and efficient strategy to identify the E3-substrate interaction. To address this challenge, we develop a computational model based on multiple types of heterogeneous biological evidence to investigate the human E3-substrate interactions. Furthermore, we provide UbiBrowser as an integrated bioinformatics platform to predict and present the proteome-wide human E3-substrate interaction network (http://ubibrowser.ncpsb.org).

[1] State Key Laboratory of Proteomics, Beijing Proteome Research Center, Beijing Institute of Radiation Medicine, National Center for Protein Sciences (The PHOENIX Center, Beijing), Beijing 102206, China. [2] Department of Biochemistry and Molecular Biology, Capital Medical University, Beijing 100069, China. [3] School of Chemistry and Chemical Engineering, Guangxi University for Nationalities, Nanning 530006, China. [4] Department of Bioinformatics and Computational Biology, The University of Texas MD Anderson Cancer Center, Houston, TX 77030, USA. Yang Li, Ping Xie and Liang Lu contributed equally to this work. Correspondence and requests for materials should be addressed to D.L. (email: lidong.bprc@foxmail.com) or to F.H. (email: hefc@nic.bmi.ac.cn)

Ubiquitin, which is an abundant 76 amino acid polypeptide, can covalently conjugate to certain proteins by an isopeptide bond between its carboxyl and the amino group of a lysine residue. This process is mediated by a cascade consistent with ubiquitin activating enzyme (E1), ubiquitin conjugating enzyme (E2), and ubiquitin ligase (E3)[1]. Ubiquitination regulates a wide spectrum of proteolytic and nonproteolytic cellular processes in eukaryotes, including proteasome-mediated protein degradation, inflammatory signaling, DNA damage response, and enzymatic activity regulation[2]. Thus, ubiquitination is closely related to the development of many diseases like Alzheimer's disease[3, 4], Parkinson's disease[5], and multiple cancers[6–8].

During the ubiquitination procedure, the interaction between ubiquitin ligase and substrate decides the substrate's specificity and determines the substrate's fate. Multiple traditional experimental strategies (e.g., GPS profiling[9], protein microarrays[10], live phage display library[11], and mass spectrometry[12]) have been developed to identify the E3-substrate interaction (ESI). However, because of E3s' low substrate levels and their intrinsically weak interactions with substrates, these methods are laborious, time intensive, expensive, and low efficient. As a result, although there are more than 30000 ubiquitin sites on over 5700 substrates in the ubiquitination site database mUbiSiDa (version 1.0)[13], only less than 900 human E3-substrate relationships are collected in database[14], which means that only a small proportion (~15%) of ubiquitinated proteins has the known corresponding ubiquitin ligase information. Therefore, a robust computational strategy is desirable to systematically identify the potential E3-substrate interaction at proteome scale.

To address this issue, we developed a naïve Bayesian classifier-based computational algorithm to combine multiple types of heterogeneous biological evidence including homology E3-substrate interaction, enriched domain and Gene Ontology (GO) term pair, protein interaction network loop, and inferred E3 recognition consensus motif, to predict human E3-substrate interactions. Then we implemented a proteome-wide E3-substrate interactions scanning generating a predicted E3-substrate interaction data set (PESID). Finally, to facilitate the usage of our algorithm and PESID, we presented an online platform (UbiBrowser) to investigate human ubiquitin ligase-substrate interaction network.

## Results

**Overview of our prediction protocol.** UbiBrowser was designed to be a naïve Bayesian classification-based platform to predict and

present human proteome-wide E3-substrate interactions (Fig. 1). First, we compiled a golden standard data set with 913 E3-substrate pairs (GESID, golden standard E3-substrate interaction data set) by manual literature mining (Papers before 1 January 2010). Then, we evaluated five types of heterogeneous evidence for the model, including homology E3-substrate interaction, enriched domain and GO term pair, protein–protein interaction network loop, and inferred E3 recognition consensus motif. We test the predictive ability of each evidence by calculating its likelihood ratio (LR). Finally, we integrated all evidence into the naïve Bayesian classification model to predict E3-substrate interactions and made the model available as web services.

**Five types of biological evidence for ESI prediction.** We used the golden standard positive (GSP) and golden standard negative (GSN) data sets to measure the reliability of each biological evidence (Methods section). For each biological evidence $f$, we calculated its LR($f$) (Fig. 2). In theory, the biological evidence $f$ with LR($f$) > 1 indicates that it has the ability to identify the true E3-substrate interaction.

E3-substrate interactions may be conserved across multiple organisms, therefore we tried to predict human E3-substrate interactions by mapping mouse E3-substrate interactions to human orthologs using the Inparanoid[15] database. Of 913 ESIs in GSP, we predicted 279(30.6%) E3-substrate interactions.

Some E3-substrate interactions are mediated by the interacting protein domains[16], therefore we thought that novel E3-substrate interactions might be predicted by identifying domain pairs enriched among known E3-substrate interactions. Domain enrichment ratio (DER) is used to assess domain pair's enrichment degree among ESIs (Methods section). We identified 3856 domain pairs that were enriched in known E3-substrate interactions, and some of them have been reported in literature. For example, the predicted E3 recognizing domain of "TP53 DNA-binding domain" was reported to interact with the E3 of WWP1[17] (Enrichment ratio: 7.21). We used two thirds of the GSP to define enriched domain pairs and the remaining to calculate LRs. We repeated this process three times and combined the results. We found that the degree of domain enrichment in GSP was strongly associated with the LR (Fig. 2a).

E3 ligases and their substrates involved in ESIs are supposed to be of the same biological functions. To test this hypothesis, we adopted the similar strategy as DER to calculate the GO term

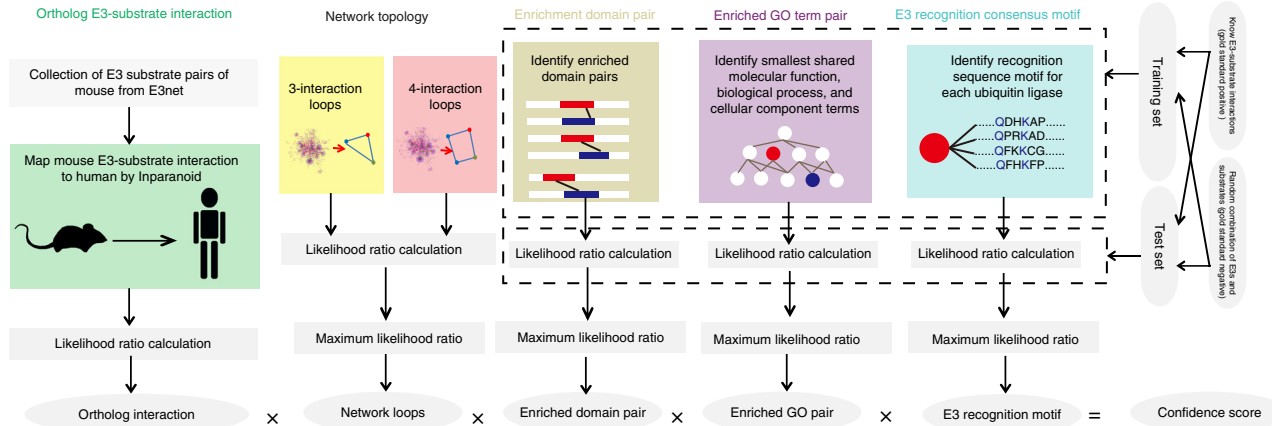

**Fig. 1** Data flow to predict the human E3-substrate interaction based on naïve Bayesian classification. Five types of evidence were labeled on the top of each data source. First, likelihood ratios were calculated as the weight for the source(s) of each type of biological evidence towards the prediction of ESI. Second, if there was more than one source for each type of biological evidence, the maximum LR from this data source was identified. Finally, the likelihood ratio from individual evidence was integrated into a composite likelihood ratio by the naïve Bayesian classifier

enrichment ratio (GER) for ESIs. We found that there is strong positive association between an ESI's GER and its LR (Fig. 2b).

Interacting proteins are always involved in certain network motifs[18]. To test whether ESI has such property, we combined the query ESI with the HPRD[19] protein interaction data to generate an integrated network. We defined $N_3$ and $N_4$ as the number of the three- and four-interaction loops that ESIs were involved, and

we found that both $N_3$ and $N_4$ can be used for prediction with satisfying performance (Fig. 2c).

E3s may bind to specific substrates by recognizing short linear sequence motifs[20, 21]. For each E3 in GSP, we predicted its recognition consensus motif based on two parallel sequence data sets: one is the sequence data of this E3's substrates in GSP to build the motif[22], and the other is that of all proteins interacting

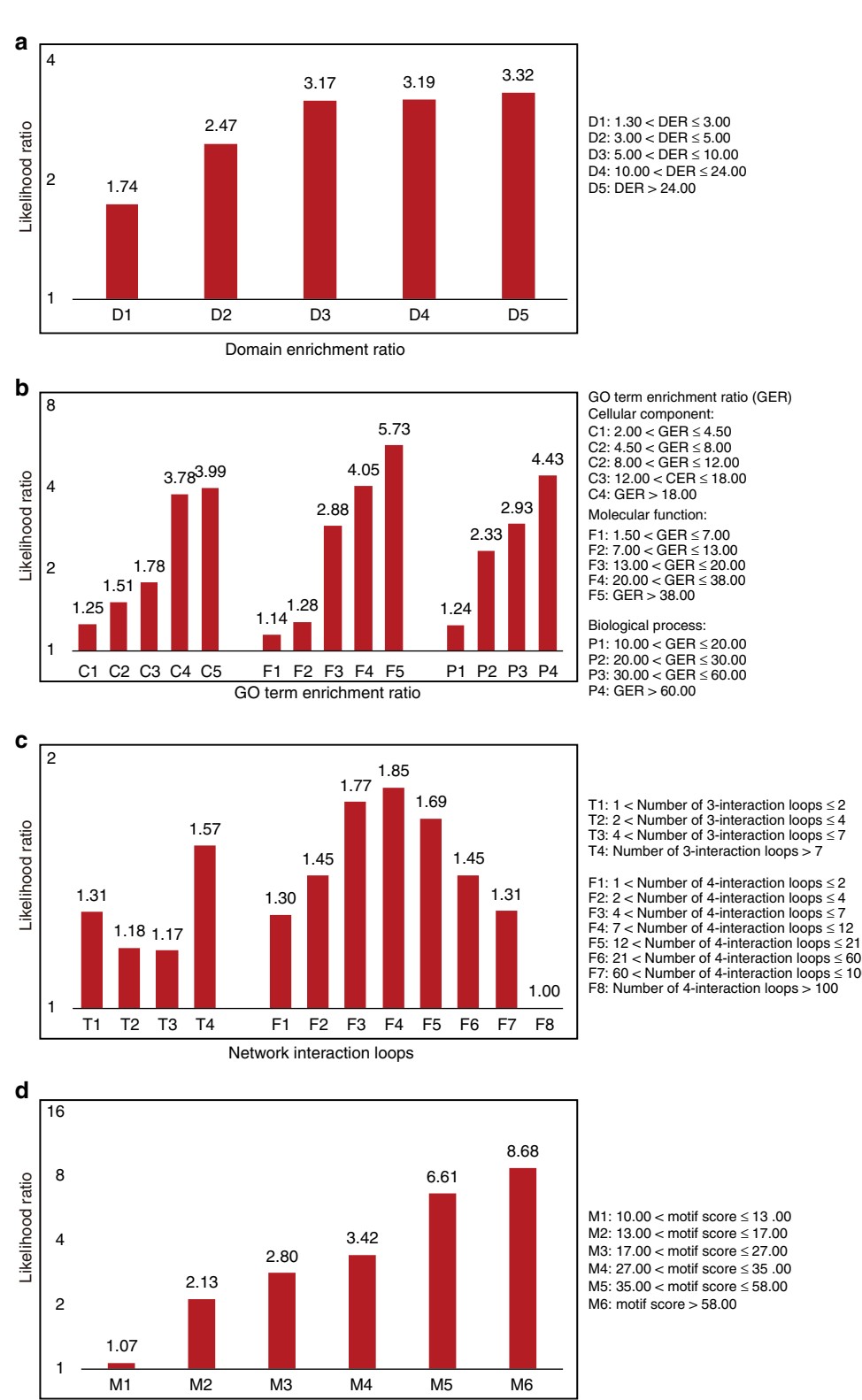

with this E3 in HPRD database for background probability calculations. We identified 10480 potential recognition consensus motifs with motif score > 2 (Supplementary Methods), these motifs were shown to be of certain prediction power (Fig. 2d), and some of them have been reported in literature. For example, "KEN" sequence motif was recognized by ubiquitin ligase complex *APC/C*[23, 24], and in our prediction results, "KEN" motif was identified as *APC/C*'s recognizing motif with the motif score: 16.13.

**The combined LR to measure the reliability of ESI**. The naïve Bayesian classification was used to integrate LRs from multiple types of data sources based on the following speculation: Bayesian classification can integrate multiple heterogeneous data sources into a common probabilistic framework and its result is easy to interpret as they represent conditional probability relationships among information sources compared to other "black-box" strategies. According to the Bayesian rules, during the prediction procedure we first identified the supporting evidence for the query ESI and assigned it the LR values. If there is more than one source for each type of biological evidence, the maximum LR is retained. And then the naïve Bayesian classifier was used to integrate these LRs from multiple types of biological evidence to generate $LR_{comp}$.

We performed a test to check whether $LR_{comp}$ can measure the reliability of an E3-substrate interaction: we changed the LR cutoff during the fivefold cross-validation against the golden standard data set, and plotted the ratio of the true to false positive (TP/FP) as the function of the cutoff of LR (Fig. 3). As shown in Fig. 3, TP/FP, acting as a measure to the accuracy of a test, increased monotonically with the cutoff of $LR_{comp}$, and this result confirmed that $LR_{comp}$ can be used as an appropriate confidence score to measure the odds of a real interaction as well as the individual LRs. Besides the Bayesian classification model, we also established the "Single Evidence Models" for each type of biological evidence, where the confidence of each E3 ligase-substrate interaction was assigned by its LR from single type of evidence.

**Performance evaluation for UbiBrowser**. Fivefold cross-validation protocol was used to evaluate the performance of our prediction model. The resulting receiver operating characteristic (ROC) curves are illustrated in Fig. 4, where each point on the ROC curves denotes the sensitivity and specificity obtained from one test against a particular $LR_{cutoff}$, and the area under ROC curve (AUROC) indicates the efficacy of the corresponding assessment system. An ideal test with perfect discrimination (100% sensitivity, 100% specificity) has an AUC 1.0, whereas a non-informative prediction has the area 0.5, indicating that it may be achieved by mere guess. The more a test's AUROC approximates to 1.0, the higher its overall efficacy will be. We found that our Bayesian model has the area of 0.827 (95% CI = 0.811–0.842) against the fivefold cross-validation, suggesting its relatively high ability to identify the ESI (Fig. 4).

Because the AUROC is an indicator of the discriminatory power for the prediction system, here we also used it to compare the efficacy of different prediction models. From Fig. 4, we noticed that those "Single Evidence Models" had different AUROCs, and that these models had significantly lower efficacy than our Bayesian model.

In addition to the fivefold cross-validation, we assessed our model on a novel independent test set of 402 ESIs that were compiled from literature after January, 2010 (none of them was used in the cross-validation for training). We found that our Bayesian model has the ROC curve area of 0.73 (95% CI = 0.697–0.769), which suggests that our model is of great prediction power for new ESIs.

**Using UbiBrowser to predict and present ESIs**. To facilitate the broad access to our ESI prediction model, we developed a user-friendly web portal-UbiBrowser. For each query of human E3 ligase or substrate protein, this portal will present the potential E3-substrate interactions in two main views: network view and sequence view (Fig. 5).

In the network view, a node is positioned in the center of canvas representing the queried E3 ligase or substrate, surrounded by the nodes representing predicted substrates or E3 ligases. In the confidence mode of network view, both edge width and node size are positively correlated with the UbiBrowser score. And in the evidence mode of network view, the edge of each predicted E3-substrate interaction is composed by multiple colored lines, with different colors representing different types of supporting evidence.

In the network view, clicking on each surrounding node will access to the corresponding sequence view of the involved E3's substrate, and in the popped sequence view, the literature-reported ubiquitination sites and predicted domains/motifs recognized by the corresponding E3 will be marked.

**Use cases of UbiBrowser**. In some pathological process, some proteins can act as the disease promoters, such as multiple oncogenes, and overexpression of these promoters will cause the occurrence and worsening of these diseases[25]. E3 ligases can regulate the stability of these disease promoters by mediating their proteasomal degradation. Detection of the E3 ligases for these disease promoters will help to reveal the underlying pathological mechanisms and further therapeutical treatments[26]. Using UbiBrowser, we tried to predict the interaction between the disease promoters and their potential upstream regulatory E3 ligases.

*ITCH-TAB1:* *TAB1* (TGF-beta-activated kinase 1-binding protein 1) can activate *p38α* (a mitogen-activated protein kinase). The activation of *p38α* can induce several inflammatory skin disorders, such as hapten-induced contact dermatitis, ultraviolet irradiation–induced dermatitis, and human psoriatic lesions[27–29]. Therefore *TAB1* was regarded as potential promoter for inflammatory skin disorders. Among the predicted E3 ligases for *TAB1* in UbiBrowser, *ITCH* ((itchy E3 ubiquitin protein

**Fig. 2** Diverse types of biological evidences contributing to the reliable evaluation. **a** Domain pairs enriched among E3-substrate interaction. DER was used to measure domain pair enrichment, which was calculated as the probability (Pr) of observing a pair of domain in a set of known E3-substrate interactions divided by the product of probabilities of observing each domain independently. **b** GO term pairs enriched among E3-substrate interaction. GER was used to measure GO term pair enrichment, which was calculated as the probability of observing a pair of GO term in a set of known E3-substrate interactions divided by the product of probabilities of observing each GO term independently. **c** Network Topology. The number of the three- and four-interaction loops was calculated based on the integrated network with the query interaction and the HPRD[19] protein interaction data. **d** E3 recognition consensus motif. Recognition consensus motif for each E3 was identified based on two parallel sequence data sets: one was the sequence data of this E3's substrates in GSP for motif building, and the other was that of all the proteins interacting with this E3 in HPRD database for background probability calculations. Please refer to Supplementary Methods for details of the calculation of E3 recognition consensus motif

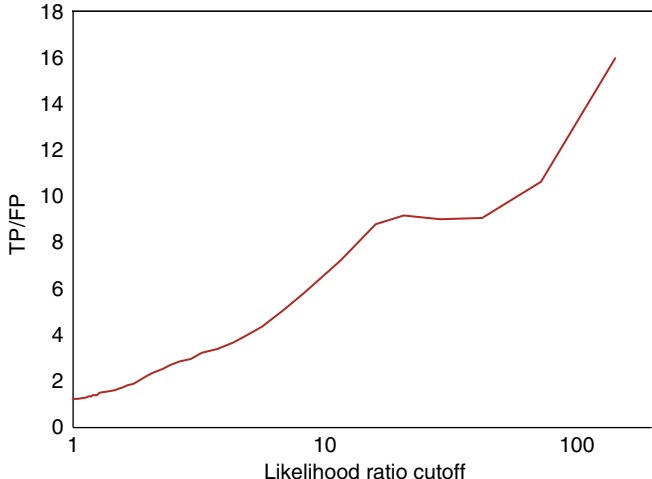

**Fig. 3** TP/FP ratio as a function of likelihood ratio cutoff for ESI prediction. The ratio of the true to false positive (TP/FP) was plotted as the function of the cutoff of likelihood ratio. The number of true positives and false positives were from the fivefold cross-validation (see text for details)

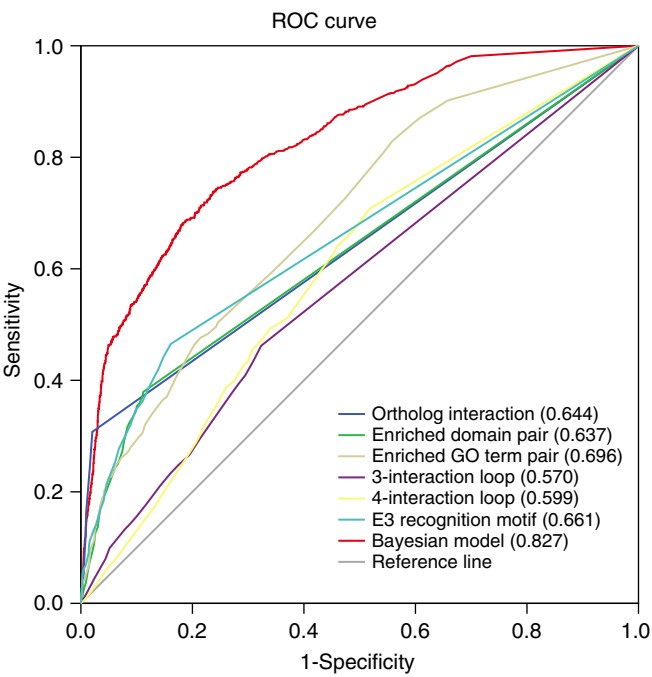

**Fig. 4** ROC curves for various assessment models using fivefold cross validations against the golden standard data sets. Each point on the ROC curves of various assessment models corresponds to sensitivity and specificity against a particular likelihood ratio cutoff. Different assessment models corresponding to these curves are labeled in legends. The numbers in parentheses refer to the AUC under ROC curves for each model. Sensitivity and specificity are computed during the fivefold cross-validations. SPSS software is used to smooth the curves. (see text for details)

ligase) was given a relative high LR: 2.65 (Rank: 12) and was reported to play an important role in inflammation and the regulation of epithelial and hematopoietic cell growth[30]. Therefore we thought that *ITCH* is the potential regulator of *TAB1*. By further investigating the supporting evidence, we found that *TAB1* matches the PPXY motif and *ITCH* always recognizes the substrate's PPXY motif using the WW domain. Our speculations about the interaction between *TAB1* and *ITCH* was validated by the recent paper[28].

*CHIP-EGFR*: *EGFR* (epidermal growth factor receptor) is the promoter of pancreatic cancer, which can initiate downstream signaling cascade, such as MAPK, PI3K/Akt, and Src pathways. *EGFR* overexpression is thought to be related with bad outcome of pancreatic cancer, thus, it is promising to inhibit the *EGFR* signaling pathway for treatment[31]. By UbiBrowser, E3 ligase CHIP (U-box dependent ubiquitin ligase) was predicted to be a potential regulator for *EGFR* (Rank:19; LR: 15.02). And this prediction can be validated by the paper published on *Oncotarget*[32], which demonstrated that *CHIP* is the E3 ligase of *EGFR*, and it might be a novel tumor suppressor in pancreatic cancer.

*NEDD4-HER3*: *HER3* ((human epidermal growth factor receptor 3) is another member of the human EGFR family. *HER3* signaling plays important roles in cell migration and proliferation in various cancers, and oncogenic *HER3* mutations have been reported in human colon and gastric cancers[33]. There are currently intense efforts toward developing anti-*HER3* antibody therapeutics for cancer treatment. We searched the potential E3 ligases for *HER3* in UbiBrowser, and we speculated that the *NEDD4* (neural precursor cell expressed, developmentally down-regulated 4) at rank 1 with a confidence score 0.908 (LR: 195.79) might be a negative regulator for *HER3*. Then by retrieving latest literature we found that NEDD4 can negatively regulate *HER3* level and signaling by mediating its ubiquitination[34].

**Experimental validation of predicted ESI**. UbiBrowser was designed primarily for assisting researchers to identify potential E3-substrate interactions. To validate whether UbiBrowser can provide potential E3-substrate interactions, we experimentally tested a pair of predicted E3-substrated interaction (Smurf1 and Smad3, LR = 29.87). We found that when co-expression of Smad3 and Smurf1, Smurf1 decreased Smad3 protein levels in a

dose-dependent manner (Fig. 6a). Then, we analyzed Smad3 in the present of ectopic wild-type (WT) Smurf1 or its ligase-inactive C699A mutant. We found that Smurf-WT significantly reduced Smad3 protein levels in MDA-MB-231 cells. In contrast, Smurf1 CA mutant hardly affected the levels of Smad3 (Fig. 6b). Because the ubiquitin ligase activity of Smurf1 was required for Smad3 degradation, we next sought to determine whether Smurf1-mediated Smad3 degradation was a consequence of ubiquitination. We used Smad1 as a positive control, which has been proved that could be ubiquitinated by Smurf1[35]. We performed the in vivo ubiquitination assay in MDA-MB-231 cells. The results showed that overexpression of Smurf1 increased the poly-ubiquitination of Smad3, but the ubiquitin chain was much weaker than Smad1 (Fig. 6c). To assess whether Smad3 interacts with Smurf1 in vivo, a co-immunoprecipitation assay was performed in MDA-MB-231 cells and the result revealed an association between Flag-Smurf1 and Myc-Smad3 in the presence of the proteasome inhibitor MG132 (Fig. 6d, e), which is consistent with Barrios-Rodiles and Ebisawa's studies, where they show there are interactions between Smurf1 and Smad3[36, 37]. As a positive control, the known ubiquitination E3 of Smad3, Smurf2 was also co-immunoprecipitated with Smad3 (Fig. 6d)[38]. To avoid the interfere of IgG Heavy Chain in immunoprecipitation, we used the anti-Myc-HRP antibody to detect Smad3. Collectively, these data indicate that under the condition of overexpression, Smurf1 functions as an E3 ligase to promote the ubiquitination and proteasomal degradation of Smad3.

**Discussion**
By analyzing and integrating five types of biological evidence we have developed a predictive model for human proteome-wide

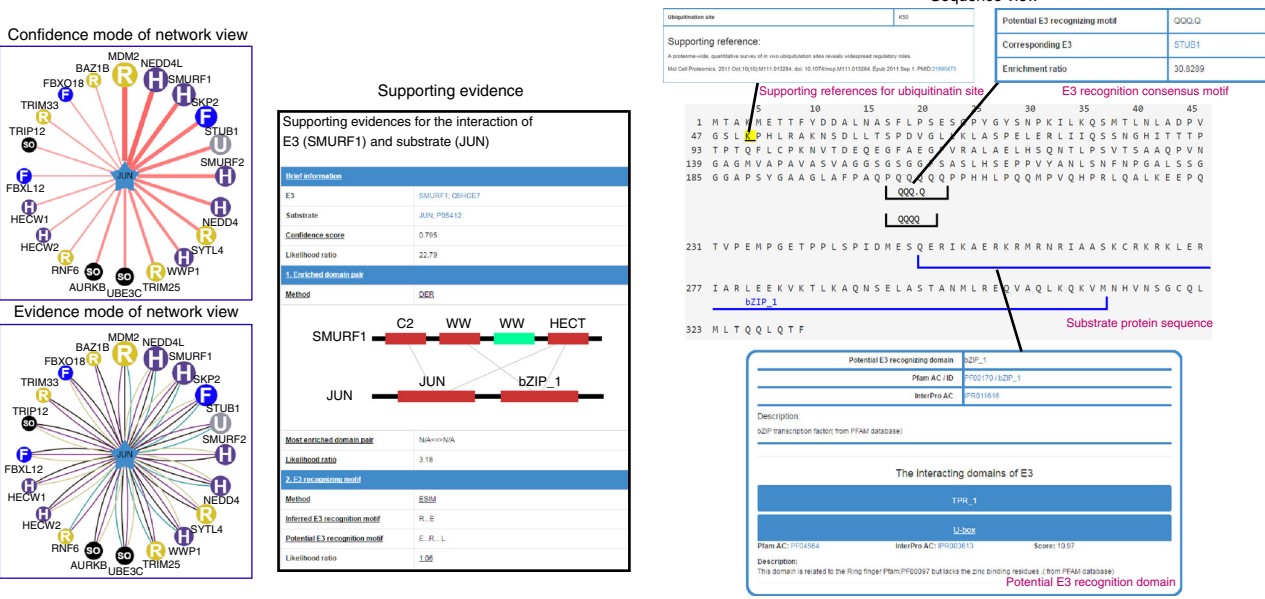

**Fig. 5** Network and sequence view for predicted E3-substrate interactions in UbiBrowser web services. In network view the central node is the queried E3 ligase or substrate, and the surrounding nodes are the predicted substrates or corresponding E3s. In the confidence mode of network view, the edge width and surrounding node size are positively correlated with the confidence of prediction, and in the evidence mode of network view, each E3-substrate interaction will be presented by multiple lines with different colors for different type of evidence. Clicking each edge will lead to a popup illustrating the supporting evidence, and clicking the surrounded node a popup for the sequence view of the involved substrate. In the sequence view for each E3-substrate interaction, the substrate's sequence is shown in PRIDE format with multiple signs: *black lines* under the sequence denote the potential domain interacting with related E3, *gray lines* under the sequence mark the inferred E3 recognition consensus motif and the *yellow background* of character K means known ubiquitination site

ubiquitin ligase-substrate relationship. We measured a total of 14,459,214 E3-protein pairs between 714 E3s and 20,251 human proteins, and identified 14,419 high confidence potential E3-substrate interactions. We also build a user-friendly E3-substrate interaction browser to present the predicted E3-substrate network together with their supporting evidence.

Kai-Yao et al.[39] have tried to construct a full protein ubiqui-tylation networks, however, their work cannot distinguish the E3's substrates and regulators. We examined 100 prediction results randomly sampling from Kai-Yao Huang et al.'s work and UbiBrowser. As shown in Supplementary Table 1, we found that only 2 results are regulators (2%) for UbiBrowser while 21 are regulators of E3s (21%) for Kai-Yao Huang et al.'s work. The ratio of regulators from UbiBrowser was significantly lower than that from Kai-Yao Huang et al.'s work ($P$ value $= 1.25 \times 10^{-5}$, one-tailed Fisher's exact test). The underlying reason is that UbiBrowser was based on a Bayesian classifier trained by the true E3-substrate interactions as GSP data sets and physical interactions between E3s and non-substrate proteins as the GSN data set, while Kai-Yao et al. only mapped the E3 ligases inter-action networks by simply incorporating experimentally verified E3 ligases, ubiquitylated substrates and protein–protein interactions.

UbiBrowser is a type of pair-input computational prediction. In this case, during the cross-validation, a test pair may share either the component with some pairs in a training set, or it may share neither. To avoid the possible over-estimation, we took the same cross-validation protocol as Park et al.[40] to divide our test data set into three parts: C1 (both E3s and substrates in the test set can be found in the training set), C2 (either E3s or substrates in the test set can be found in the training set), and C3 (neither E3s nor substrates in the test set can be found in the training set). We found that the AUROC against C1 is 0.855 (95% CI = 0.833–0.876, Supplementary Fig. 1). Interestingly, compared

to the results in Park et al.'s paper, in the case of C2 and C3, although there exist E3s or substrates that do not appear in the training data set, we found that our prediction system has certain prediction power (AUROC against C2 and C3: 0.816, 95% CI = 0.794–0.837 and 0.629, 95% CI = 0.468–0.790). The reason is that UbiBrowser can utilize the underlying domain, motif, and GO features for proteins even these proteins are absent from the training data sets.

Yamashita et al. found that the protein abundances of Smad (1,2,3,5) were not influenced in *Smurf1*-/- mice, but they also agreed that BMP pathway and TGF-β pathway were regulated in cells overexpressing Smurf1[41]. In our experiment, we found that Smad3 could interact with Smurf1, and Smurf1-mediated Smad3 ubiquitination in overexpression condition (Fig. 6c–e). Several studies are also available for the interactions between Smurf1 and Smad3. For example, Miriam Barrios-Rodiles's[42] study showed the PPI between Smurf1 and Smad3 can be identified by high-throughput methods. Takanori Ebisawa's study[43] showed that there is weak interaction between Smurf1 and Smad3. Although Smad3 ubiquitination reduction is not obvious when in vivo Smurf1 was knocked out[41], we considered that ubiquitination of Smad3 is compensated by other ubiquitin ligases, such as Smurf2. Previous study has reported that Smad3 is a major substrate of Smurf2-mediated ubiquitination[44, 45]. It has been suggested that Smurf2 may partially compensate the function for the loss of Smurf1. *Smurf1*−/− and *Smurf2*−/− (Smurf DKO) mice display embryonic lethality at around E12.5. However, single Smurf1 or Smurf2 mice have no overt defects in embryogenesis 4. Therefore the in vivo ubiquitination level change of Smad3 is difficult to be detect.

## Methods

**Golden standard positive data sets**. Abstracts before 1 January 2010 were downloaded from PubMed using key words "ubiquitin ligase". All these abstracts

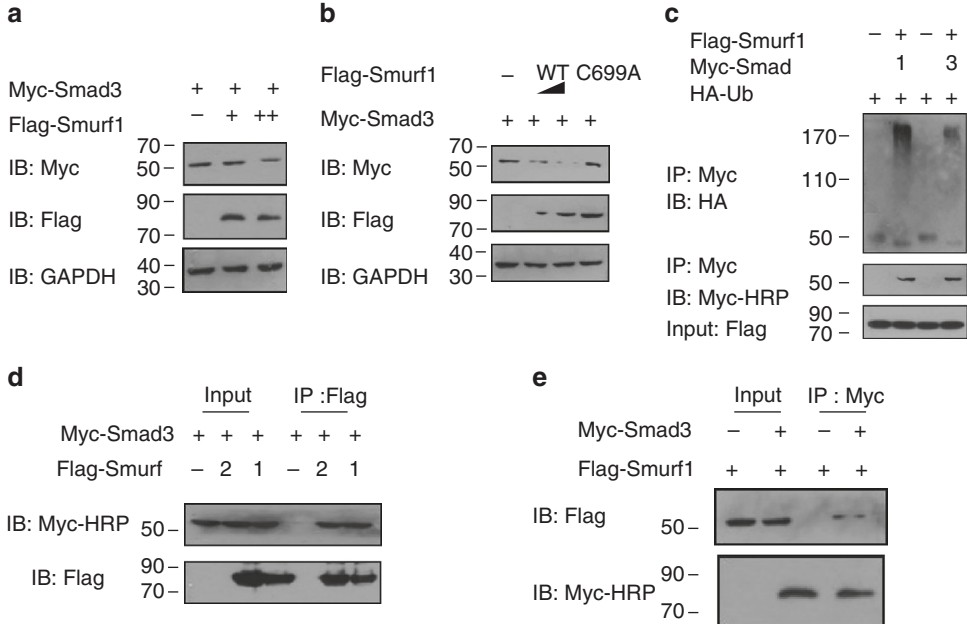

**Fig. 6** Experimental validation of predicted E3-substrate interaction. **a**, **b** Smurf1 destabilizes Smad3 in MDA-MB-231 cells. **a** MDA-MB-231 cells were transfected with Myc-Smad3 and Smurf1, after 36 h, cells lysates were analyzed by western blot. **b** MDA-MB-231 cells were transfected Myc-Smad3, with Flag-Smurf1 (WT or C699A: 0.5 μg of Flag-Smurf1-WT was transfected into the second band and 1 μg into the third one, and 1 μg of Flag-Smurf1-C699A into the fourth one). Myc-Smad3 level was analyzed by immunoblotting. **c** Smurf1 promotes the ubiquitination of Smad3. MDA-MB-231 cells were transfected with HA-Ub, Myc-Smad3, control vector, or Flag-Smurf1, Smad1 was used as a positive control, and treated with MG132 as indicated. Ubiquitinated Smad3 was immunoprecipitated (IP) with anti-Myc antibody and detected by immuneblotting with anti-HA antibody. **d**, **e** Smurf1 interacts with Smad3. Co-immunoprecipitation of Smurf1 and Smad3 in MDA-MB-231 cells. In Fig. 6d, Flag-Smurf1 was used as immunoprecipated, and in Fig. 6e, Myc-Smad3 was used as immunoprecipated. To avoid the degradation of Smad3, MG132 (20 μM) was added for 8 h before harvested. Cell lysates were immunoprecipitated with anti-Myc antibody and analyzed by immunoblotting

were sent to "E3miner" (a text-mining tool) to extract the potential E3-substrate interactions. Then the potential E3-substrate interactions were manually filtered based on the following patterns: "E ubiquitylates S…", "E mediate the ubiquitination of S…", "E target S for ubiquitination…", "E promote the proteasome degradation of S…", "E target S for degradation…", "E promote the ubiquitination of S…", "E plays a crucial role in the ubiquitination of S…", "S is the substrate of E…", "S is ubiquitinated and degraded by E…", or "S is resistant to degradation mediated by E…", where E is an E3 and S is an E3 ligase substrate. Finally, 913 E3-substrate interactions were manually extracted from these published papers to construct the GSP for cross-validation. Following the same literature mining protocol, we also obtained a novel independent test set of 402 ESIs from literature after 1 January 2010 to test whether UbiBrowser has the ability to predict novel ESIs. All these E3-substrate interactions data set together with the supporting literature information were provided as Supplementary Data 1.

**Golden standard negative data set**. It is difficult to find an experimental negative data set, therefore, we built our GSN data set based on protein physical interaction data set. The physical protein interaction data sets were downloaded from HPRD[19] (Released 13 September 2005), IntAct[46] (Released 5 June 2013), and IrefIndex[47] (Released 9 December 2013). These data sets were integrated into a non-redundancy protein interaction data set for E3s containing 10109 physical interactions. After random screening, we obtained 2734 results as the GSN data set (none of them were in GSP and literature).

**Construction of multiple types of biological evidence**. 366 pairs of mouse E3-substrate interactions were collected from E3Net[14] database. Pairwise ortholog map file ("mouse to human") were downloaded from the Inparanoid[15] database (Released 8.0).

Protein domain and family assignments data sets were downloaded from Pfam[48] (Released version 27). In total, 45019 assignments of 5487 protein domains and families to one or more of 18312 proteins were retrieved. Domain pair enrichment was assessed with the DER[49], which is calculated as the probability (Pr) of observing a pair of domain in a set of known E3-substrate interactions divided by the product of probabilities of observing each domain pair independently:

$$DER = \Pr(d_{e3}:d_{sub}|GSP) \div (\Pr(d_{e3}|GSP) \times \Pr(d_{sub}|GSP)) \quad (1)$$

where $d_{e3}$ is a domain of E3, $d_{sub}$ is a domain of substrate, and $d_{e3}:d_{sub}$ is an E3-substrate interaction in which E3 has $d_{e3}$ and substrate has $d_{sub}$.

GO annotation data was downloaded from Gene Ontology Consortium (http://geneontology.org, Released on 27 November 2014). GO term pair enrichment was assessed with the GER, which is calculated as the probability (Pr) of observing a pair of GO terms in a set of known E3-substrate interactions divided by the product of probabilities of observing each GO term independently:

$$GER = \Pr(g_{e3}:g_{sub}|GSP) \div (\Pr(g_{e3}|GSP) \times \Pr(g_{sub}|GSP)) \quad (2)$$

where $g_{e3}$ is a GO term of E3, $g_{sub}$ is a GO term of substrate, and $g_{e3}:g_{sub}$ is an E3-substrate interaction in which E3 has $g_{e3}$ and substrate has $g_{sub}$, and GSP is a GSP set of known E3-substrate interaction.

E3 recognition consensus motif was calculated based on two parallel sequence data sets: one is the target data set containing sequences of this E3's substrates in GSP, the other is background data set those of all proteins in GSP which interact with this E3 in HPRD[19] for background probability calculations. The sequence of human proteome was downloaded from Swiss-Prot. E3 recognition consensus motif was identified by modified procedure from motif-x[22] (Supplementary Methods).

**Bayesian models for prediction**. Here, we defined interactions that ESI occurs as "positive" and those that does not as "negative". The prior odds of an interaction that ESI occurs can be calculated based on the ratio of the probability of detecting a pair of ESI from all protein pairs (estimated by the golden standard data sets):

$$O_{prior} = P(Positive) \div P(Negative)) = P(Positive) \div (1 - P(Positive)) \quad (3)$$

And the posterior odds ($O_{post}$) of an interaction that ESI occurs is defined as:

$$O_{post} = P(Positive|f) \div P(Negative|f) \quad (4)$$

where $P(Positive|f)$ is the probability that an ESI occurs after considering the biological evidence $f$, while $P(Negative|f)$ stands for the possibility that it does not. Following a derivation of Bayesian rules[50], the posterior odds ($O_{post}$) of an interaction that ESI occurs can be calculated as the product of the prior odds ($O_{prior}$) and the LR($f$) by equation:

$$O_{post} = O_{prior} \times LR(f) \quad (5)$$

where LR($f$) is the ratio of the probability of meeting condition $f$ of interacting ESI pair and non-interacting ESI pair in the golden standard data sets. From Eqs. 3 and 5, the LR for biological evidence $f$ can be computed as:

$$LR(f) = P(f|Positive) \div P(f|Negative) = \frac{TP_f/T}{FP_f/F} \quad (6)$$

where $T$ and $F$ are the number of all the true and false interactions respectively, $TP_f$ and $FP_f$ are the number of true and false interactions with the biological evidence $f$ respectively. The advantages of Bayesian rules in this system permit us to integrate multiple heterogeneous data sources into a probabilistic model. Because these biological data types integrated are obtained by different approaches, we assume that they are conditionally independent. Therefore, we can get the composite LR ($LR_{comp}$) by simply multiplying the LRs from individual sources, which is namely the naïve Bayes classification (Eq. 7).

$$LR(f_1...f_n) = \prod_{i=1}^{i=n} \left( P(f_i|Positive) \div P(f_i|Negative) \right) = \prod_{i=1}^{i=n} LR(f_i) \quad (7)$$

According to the Bayesian rules described above, during the prediction procedure we first identified the supporting evidence for the query ESI and assign it the LR values. If the biological evidence in the same data type give more than one LR, the maximum will be retained. And then the naïve Bayesian classifier was used to integrate these LRs from multiple types of data sources to generate $LR_{comp}$, which was further normalized into "UbiBrowser Score" for confidence assessment.

$$UbiBrowser\ Score = \frac{1}{1 + e^{-\log\left(LR_{comp}\right)}} \quad (8)$$

**ROC curve and cross-validation**. ROC curve can show the efficacy of one test by presenting both sensitivity and specificity for different cutoff points[51]. Sensitivity and specificity can measure a test's ability to identify true positives and false positives in a data set. These two features can be calculated as Sensitivity = $(TP)/(T)$ and Specificity = $1-(FP)/(F)$, where TP and FP are the number of identified true and false positives, while $T$ and $F$ represent the total number of positives and negatives in a test. The ROC curves were plotted and smoothed by SPSS software with the "sensitivity" on the y-axis and "1-specificity" on the x-axis.

To test the efficacy of the overall performance of various assessment models, the fivefold cross-validation protocol was used. The GSP and negative data sets were randomly divided into five approximately equal subsets. Four sets were used as training data sets to compute the individual evidence's LRs. The remaining one was used as the test data set to count the number of predicted true positive (TP) and false positive (FP) where one protein pair is predicted to be positive if its LR exceeds a particular cutoff, $LR_{cutoff}$, and to be negative otherwise. This process was done in turn five times, and finally the numbers of TPs and FPs against different LRs across five test data sets were summed to calculate the TP/FP ratio, and the sensitivity (TP/$T$) and specificity (1-FP/$F$) for the ROC curve.

**Implementation of UbiBrowser web services**. UbiBrowser web services were constructed based on a MySQL database, which was designed to be a general database for storing the predicted ESI and their annotations. Above this database, analysis applications were implemented in PHP and Perl for processing, integrating and indexing the data, and web presentation application were written in JavaScript and CSS (Cascaded Style Sheets). UbiBrowser is currently running on a single 4-CPU Ubuntu Linux server, with the Apache 2 HTTP server.

**Experimental validation protocols**. Full-length of Smad3 and Smurf1 were constructed by PCR, followed by subcloning into various vectors. Anti-Smad3 (Cat# 9523, 1:1000)[52], anti-Myc (Cat# 2276,1:2000)[53] and anti-Myc-HRP (Cat# 2040, 1:1000)[54] was from Cell Signaling Technology. Anti-Smurf1 (Cat# ab117552 1:1000)[55] was from Abcam. Anti-Flag M2 monoclonal antibody (Cat# F3165, 1:2000)[56] and the proteasome inhibitor MG132 (Cat# M8699)[57] were from Sigma Aldrich. Anti-HA antibody was from Roche Life Science (Cat# 11867423001, 1:2000). GAPDH (Cat# sc-47724, 1:2000)[58] and secondary antibodies (goat anti-rabbit-HRP Cat# sc-2030, 1:3000; goat anti-mouse-HRP Cat# sc-2005, 1:3000)[59] were purchased from Santa Cruz Biotechnology. The MDA-MB-231 cell line was kindly gifted by professor Wenguo Jiang from Cardiff University School of Medicine. These cell lines were authenticated by STR locus analysis (Genetic Testing Biotechnology Corporation, Suzhou, China) and tested for mycoplasma contamination. Cell was cultured in DMEM/F12 medium (Hyclone). These cells were supplemented with 10% fetal bovine serum (Hyclone), penicillin (50 U/ml), and streptomycin (50 lg/ml) (Hyclone). Cells were transfected with Lipofectamine 2000 (Invitrogen) according to the manufacturers' instructions. Cells were harvested and lysed in HEPES lysis buffer (20 mM HEPES pH 7.2, 50 mM NaCl, 0.5% Triton X-100, 1 mM NaF, 1 mM dithiothreitol) supplemented with protease inhibitor cocktail (Roche Life Science, Cat# 04693116001). The lysate was incubated with indicated antibody 3 h at 4 °C, then added protein A/G-plus

agarose and rotated gently more than 8 h at 4 °C. The immunoprecipitates were washed at least three times in lysis buffer, and analyzed by western blotting.

In vivo modification assays: To prepare cell lysates, cells were solubilized in modified lysis buffer (50 mM Tris, pH 7.4, 150 mM NaCl, 10% glycerol, 1 mM EDTA, 1 mM EGTA, 1% sodium dodecyl sulfate (SDS), 1 mM Na3VO4, 1 mM DTT, and 10 mM NaF) supplemented with a protease inhibitor cocktail. The cell lysate was incubated at 60 °C for 10 min. The lysate was then diluted 10 times with modified lysis buffer without SDS. The lysate was incubated with the indicated antibody for 3 h at 4 °C. Protein A/G-plus Agarose was added, and the lysate was rotated gently for 8 h at 4 °C. The immunoprecipitates were washed at least three times in wash buffer (50 mM Tris, pH 7.4, 150 mM NaCl, 10% glycerol, 1 mM EDTA, 1 mM EGTA, 0.1% SDS, 1 mM DTT, and 10 mM NaF). Proteins were recovered by boiling the beads in sample buffer and analyzed by western blot analysis.

**Statistical analyses**. All statistical analyses were performed using SPSS (version 20). ROCs curves were plotted and smoothed, and the area under the curve (AUROC) and its 95% confidence interval was simultaneously calculated. To determine if there are nonrandom associations between two categorical variables, statistical significance was considered at $P < 0.05$ using the one-tailed Fisher's exact test. Unless otherwise stated, all experiments were repeated at least thrice ($n = 3$).

**Data Availability**. All relevant data and codes are available from the authors upon request.

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

## Acknowledgements

We would like to thank Jun Qin, Ping Xu, Yingxian Li, and Pei Zhen for their fruitful discussion. We would like to thank Beijing Genestone Technology Ltd for their support on the website development. This work is funded by the Program of International S&T Cooperation (2014DFB30020), Program of Precision Medicine (2016YFC0901905), National Natural Science Foundation of China (31271407), Chinese High Technology Research and Development (2015AA020108) and U.S. National Institutes of Health (CA175486 and CA209851).

## Author contributions

D.L., F.H., and H.L. conceived and supervised the project. Y.L., L.L., and D.L. implemented the system and designed the UbiBrowser web site. P.X. implemented the experimental validation. L.D., Q.H., D.L., and Y.L. implemented the UbiBrowser web site. Z.L., F.G., Y.H, J.W., and Y.L. participated in the analyses. All authors read and approved the final manuscript.

## Additional information

**Competing interests:** The authors declare no competing financial interests.

