## [Peer Review file · Nature Communications]

Reviewers' comments:

Reviewer #1 (Remarks to the Author):

This manuscript describes a new tool called Unibrowrser, which is a predictor of E3-substrate interactions. It is somehow a novel idea, although there were some attempts towards mapping of interaction networks of E3 ligases e.g. Huang, Kai-Yao, et al. "A new scheme to discover functional associations and regulatory networks of E3 ubiquitin ligases." BMC Systems Biology 10.1 (2016): 27.

The authors should discuss that.

In my opinion the main issues with this manuscript are the following:

1. The authors do not explain in Fig 1 how much weight each of the components have towards the prediction of E3 ligase substrates.
2. The authors used the tool to study set of 30 ESIs compiled from literature after their model was built. Only 12 out of 30 ESIs had a likelihood ratio of over 10. Discussion with explanations should follow immediately as without it the reader is left to believe that the model might not be predictive.
3. The authors did not really predict any new interactions followed by biochemical characterization, which would be ideal if done in collaboration with another lab, although the section "Use case of Unibrowrser" describes some data from literature.

As a minor point, the authors should fix any grammatical mistakes, which are present throughout the manuscript.

Reviewer #2 (Remarks to the Author):

In this manuscript, a novel method to predict E3 ubiquitin ligase substrates is discussed and implemented. In addition, a web-server is presented that visualizes prediction results. The method uses a Bayesian formulation and integrates multiple sources of biological evidence to make predictions on whether a protein is a substrate for a specific ligase or not. The browser is designed and implemented well and non-technical users will find it easy to use it. The authors claim that the performance of this approach is superior to using each piece of biological evidence in isolation. However, the evaluation protocols described in this work do not appear to be rigorous enough to be convincing. Furthermore, improvements in presentation and writing would be necessary.

Major:

- the "Methods" section is not detailed enough for one to reproduce the work. For example, how were the E3-substrates in the positive set collected manually? What were the search approaches used and what sources of literature were heavily relied on? Were there any specific criteria for curators to claim a protein to be an E3 substrate?
- by definition, if a protein is a substrate for an enzyme, the two should interact with each other. Isn't the use of physical interactions as a feature too obvious, then? If the negative set is simply a random sample of ligase-protein pairs, then this feature will be a strong predictor. Interestingly, the authors observe that ~43% of the positive set overlap with the physical interactions data set. Can they explain this low number in the manuscript? Is it more likely that their interaction database is incomplete or (more importantly) that their positive set contains many spurious E3-

substrate relationships?

- on a related note, the choice of a negative set provides a distinct advantage to the predictor because a ligase and a random protein are less likely to interact with each other. It is suggested that additional, more stringent negative sets be constructed - (1) a random sample of known interacting protein pairs, (2) a random sample of enzyme-substrate pairs (e.g. other ligases or kinases with their substrates). In other words, the negative set should ideally come from the space of known protein-protein interactions (PPIs) or known enzyme-substrate interactions.

- another important issue has to do with the cross-validation procedure. In these types of problems, one has to make partitions carefully to avoid overestimation of performance. Generally, on prediction problems that concern pairs of objects, features from both members tend to be encoded (in this case, the domains and GO terms). It has been suggested if a pair occurs in the training set, neither of its members should be involved in any pairs in the test set (see Park and Marcotte. "Flaws in evaluation schemes for pair-input computational predictions" *Nature Methods*, 2012). Specific to this case, to ensure that performance is not overestimated, it may at least be useful to have all E3-substrate pairs for a given ligase be entirely in the training set in each fold. It appears that this is not the approach taken in the manuscript.

- although there may not be methods to compare the given approach to, this problem can be treated as a special case of the PPI prediction problem. There are several methods in this domain and it is recommended to compare and contrast the UbiBrowser approach to some of these methods. One interesting method is FpClass (Kotlyar et al. "In silico prediction of physical protein interactions and characterization of interactome orphans" *Nature Methods*, 2015) as it uses the same principle of evidence integration as this work.

Minor:

- in the "Six types of..." sub-section in the Results section, the term abbreviation GSN is introduced without any previous explanation of what it stands for.

- there is a typographical error in equation 3: the second step is missing $P(\text{positive}) / P(\text{negative})$; i.e., the term for Oprior.

- it is not clear what the motivation for the UbiBrowser score formula is and this should be explained in the text. Why was this transformation chosen?

- although it is not necessary, a discussion section has become the norm in most scientific papers and it is suggested that a section be devoted to discussing some of the results, especially in the context of related work and previous literature.

Reviewer 1

Q1: This manuscript describes a new tool called Ubibrowser, which is a predictor of E3-substrate interactions. It is somehow a novel idea, although there were some attempts towards mapping of interaction networks of E3 ligases e.g. Huang, Kai-Yao, et al. "A new scheme to discover functional associations and regulatory networks of E3 ubiquitin ligases." *BMC Systems Biology* 10.1 (2016): 27. The authors should discuss that.

A: Thanks for your great suggestion.

Yes, UbiBrowser is the first predictor for E3-substrate interaction although Huang, Kai-Yao, et al. have tried to map the interaction networks of E3 ligases. There are significantly differences.

1) Motivation:

Kai-Yao Huang et al.'s work only aimed to provide "a full investigation of protein ubiquitylation networks by incorporating experimentally verified E3 ligases, ubiquitylated substrates and protein-protein interactions (PPIs)".

Therefore, their work can not distinguish the E3's substrates and regulators.

Estimation based on our manual curation for their results (Table S2), only a small proportion (12.5%) of the interactions is E3-substrate interactions.

UbiBrowser was specially designed for the prediction of E3-substrate interactions. For this, true E3-substrate interactions were used as golden standard positive datasets and physical interactions between E3s and non-substrate proteins as the golden standard negative dataset for training. And 5 efficient biological evidences were used for prediction. UbiBrowser can efficiently distinguish the E3's substrates and regulators (ROC AUC=0.838, Figure 4). We also performed the experimental validation for our UbiBrowser prediction (Figure 6).

2) Manual curation:

To better show the above difference, a systematic manual curation was performed by us. We randomly chose 100 results from UbiBrowser and 100 results from Kai-Yao Huang et al. Then, we manually checked the above 200 pairs to

investigate the proportion of the regulatory proteins. As shown in Table S3, we found that only 2 results are regulators of E3s (2%) for UbiBrowser while 21 are regulators (21%) for Kai-Yao Huang et al.'s work. The ratio of regulators from UbiBrowser was significantly lower than that from Kai-Yao Huang et al.'s work (P value= 1.25×10^{-5} , Fisher's exact test).

Q2: The authors do not explain in Fig 1 how much weight each of the components have towards the prediction of E3 ligase substrates.

A: Thanks for your suggestion.

In Figure 1, the weight of each component is measured by the likelihood ratio. Our prediction system is established based on the Bayesian rule. Likelihood ratios are used to measure the prediction efficacy of biological features. If there is more than one source for each biological evidence, the maximum likelihood ratio is retained. And then the naïve Bayesian network is used to integrate these *LRs* from multiple types of data sources to generate LR_{comp} , which is further normalized into “UbiBrowser Score” for confidence assessment.

We have revised the Figure legend for Figure 1. Please see line 539 for details.

Q3: The authors used the tool to study set of 30 ESIs compiled from literature after their model was built. Only 12 out of 30 ESIs had a likelihood ratio of over 10. Discussion with explanations should follow immediately as without it the reader is left to believe that the model might not be predictive.

A: Thank you for your suggestion. We have added the related discussion for this issue.

In our previous manuscript, 12 of 30 ESIs compiled from literature had a predicted likelihood ratio larger than 10. This result doesn't mean that our model is not

predictive. In fact, compared to the random selection, this result has the enrichment ratio of 32.4. In addition, if users set the cutoff as 1, 27 of 30 E3s will be predicted as positive (we set the cutoff as 10 to achieve a relative high accuracy).

In our revised manuscript, according to reviewer2's suggestion, we optimized our model and deleted the feature of physical interaction.

To better show the efficacy of our system, an independent positive dataset containing 301 E3-substrate interactions are used to test our model (physical interaction dataset as negative). The AUROC of our improved model is 0.749 against this independent dataset, which means that our model has a satisfied efficiency.

Q4: The authors did not really predict any new interactions followed by biochemical characterization, which would be ideal if done in collaboration with another lab, although the section "Use case of Unibrowser" describes some data from literature.

A: Thanks your great suggestion. In fact, we have collaborated with our colleagues for the experimental validations. We have successfully validated a new E3-substrate interaction (Smurf1 and Smad3, $LR=406.01$) and we have added this result in our revision manuscript.

Please refer to Figure 6 and line 227 for details.

Reviewer 2

Q1: the "Methods" section is not detailed enough for one to reproduce the work. For example, how were the E3-substrates in the positive set collected manually? What were the search approaches used and what sources of literature were heavily relied on? Were there any specific criteria for curators to claim a protein to be an E3 substrate?

A: Many thanks for your suggestion.

In the revised manuscript, as you suggested, we have added more details for the "Methods" section.

As for the issue of “the E3-substrates in the positive set collected manually”, the procedure is:

- 1) We downloaded abstracts before January 1, 2010 from PubMed database using key words “ubiquitin ligase”.
- 2) We sent all these abstracts to a text-mining tool to extract the potential E3-substrate interactions. [Lee, Hodong, Gwan-Su Yi, and Jong C. Park. Nucleic acids research 36.suppl 2 (2008): W416-W422.]
- 3) Then we manually checked these potential E3-substrate interactions with strict criteria.

As for the issue of “any specific criteria for curators to claim a protein to be an E3 substrate”:

The claimed human E3 substrate (S) and its E3 (E) must meet one of the following patterns:

“E ubiquitylates S...”

“E mediates the ubiquitination of S...”

“E targets S for ubiquitination...”

“E promotes the proteasome degradation of S...”

“E targets S for degradation...”

“E promotes the ubiquitination of S...”

“E plays a crucial role in the ubiquitination of S...”

“S is the substrate of E...”

“S is ubiquitinated and degraded by E...”

“S is resistant to degradation mediated by E...”

We have presented all these E3-substrate interactions dataset together with the supporting literature information as supplementary material (Supplementary Data 1).

And as for other issues for “**Experimental validation of predicted E3-substrate interaction**”, “**Golden standard negative dataset**” and “**Bayesian models for prediction**”, we also provided more details. Please refer to line 227, line 283, line 343 for details.

Q2-1: by definition, if a protein is a substrate for an enzyme, the two should interact with each other. Isn't the use of physical interactions as a feature too obvious, then? If the negative set is simply a random sample of ligase-protein pairs, then this feature will be a strong predictor.

A: Many thanks for your comments.

Yes, there is strong correlation between physical interaction and E3-substrate interaction.

According to your suggestion in Q3, we deleted the feature of physical interaction from our prediction system, and used the physical interactions between E3s and non-substrate proteins as the golden standard negative dataset. The revised model also has satisfied performance (Figure 4).

Q2-2: Interestingly, the authors observe that ~43% of the positive set overlap with the physical interactions data set. Can they explain this low number in the manuscript? Is it more likely that their interaction database is incomplete or (more importantly) that their positive set contains many spurious E3-substrate relationships?

A: Thanks for your comments.

The reason of the low number of overlap between the positive set overlap with the physical interactions data set is that these two datasets were collected from different resource.

The golden standard positive E3-substrate dataset was from literature mining while physical interaction dataset is from PPI databases.

The golden standard positive dataset is obtained by manual curation with very strict criteria (Please refer to the answer to Q1 for details), and is of high confidence. We think that most of these ESIs in literature have not been deposited in the PPI database because of their delay in text mining.

Q3: on a related note, the choice of a negative set provides a distinct advantage to the predictor because a ligase and a random protein are less likely to interact with each other. It is suggested that additional, more stringent negative sets be constructed - (1) a random sample of known interacting protein pairs, (2) a random sample of enzyme-substrate pairs (e.g. other ligases or kinases with their substrates). In other words, the negative set should ideally come from the space of known protein-protein interactions (PPIs) or known enzyme-substrate interactions.

A: Thanks for your great suggestion.

As you suggested, we have reconstructed the golden standard negative dataset (GSN) from the space of known protein interactions database.

The detail of the GSN construction is:

Step 1: We obtained the list of E3 from both the database of UUCD and the golden standard positive datasets we constructed.

Step 2: We retrieved the interacting proteins for the E3s obtained in Step 1 from HPRD, IntAct and IRefIndex. All the interactions in golden standard positive are excluded.

Step 3: We randomly select 1022 interactions from the retrieval of Step 2 to balance the GSP and GSN.

Based on the new GSN, 5-fold cross validation and independent test were used for evaluation. The corresponding area under ROC is 0.838 and 0.749 respectively, which means that our model has satisfied performance.

Q4: another important issue has to do with the cross-validation procedure. In these types of problems, one has to make partitions carefully to avoid overestimation of performance. Generally, on prediction problems that concern pairs of objects, features from both members tend to be encoded (in this case, the domains and GO terms). It has been suggested if a pair occurs in the training set, neither of its members should be involved in any pairs in the test set (see Park and Marcotte. "Flaws in evaluation schemes for pair-input computational predictions" Nature Methods, 2012). Specific to

this case, to ensure that performance is not overestimated, it may at least be useful to have all E3-substrate pairs for a given ligase be entirely in the training set in each fold. It appears that this is not the approach taken in the manuscript.

A: Thanks for your great suggestion.

To avoid the possible overestimation of performance, we followed the protocol of Park and Marcotte, and divided our test dataset into three parts: C1 (Both E3s and substrates in the test set are found in the training set), C2 (Either E3s or substrates in the test set are found in the training set) and C3 (Neither E3s nor substrates in the test set are found in the training set).

The AUROC against C1, C2 and C3 is 0.842, 0.844 and 0.684, respectively.

Interestingly, compared to the results in Park and Marcotte's paper, in the case of C2 and C3, although there exist E3s or substrates that don't appear in the training dataset, our prediction system can also predict the true E3-substrate interactions for them.

Q5: although there may not be methods to compare the given approach to, this problem can be treated as a special case of the PPI prediction problem. There are several methods in this domain and it is recommended to compare and contrast the UbiBrowser approach to some of these methods. One interesting method is FpClass (Kotlyar et al. "In silico prediction of physical protein interactions and characterization of interactome orphans" Nature Methods, 2015) as it uses the same principle of evidence integration as this work.

A: Thanks for your suggestion.

According to your suggestion in Q3, we took the non-E3-substrate physical interactions as golden standard negative dataset. Therefore, both the interactions in GSP and GSN are physical interaction. Obviously, the traditional PPI prediction model cannot distinguish the interactions in GSP and GSN.

To support the above speculation, we have compared our model with STRING database based on the golden standard positive dataset (manually checked

E3-substrate interaction) and golden standard negative dataset (non E3-sbustrate protein-protein interaction). The AUROC of our model is 0.749 while that of String is 0.524, which means that our model can distinguish the E3-substrate interaction efficiently while the traditional method can not.

Q6: in the "Six types of..." sub-section in the Results section, the term abbreviation GSN is introduced without any previous explanation of what it stands for.

A: Thanks for your suggestion.

GSN is the abbreviation for "Golden Standard Negative".

We have added the explanation for GSN in the "**Five types of biological evidences to predict the E3-substrate interactions**" sub-section in the Results section.

Sorry for the mistake.

Q7: - there is a typographical error in equation 3: the second step is missing $P(\text{positive}) / P(\text{negative})$; i.e., the term for O_{prior} .

A: Thanks for your reminding.

And we have rewritten the related sections.

Please see line 343 for details.

Q8 : it is not clear what the motivation for the UbiBrowser score formula is and this should be explained in the text. Why was this transformation chosen?

A: Thanks for your comments.

The motivation of this transformation is to provide user an intuitionistic score rather than the likelihood ratio itself.

And this score is a logistic transformation, [Hosmer Jr, David W., and Stanley Lemeshow. Applied logistic regression. John Wiley & Sons, 2004.] which has been

used for the problem of generation of bounded outcome scores. [Lesaffre, Emmanuel, Dimitris Rizopoulos, and Roula Tsonaka. *Biostatistics* **8**, 72-85(2007).]

According to reviewers' recommendations, we have added the above speculations in the discussion section.

Reviewers' comments:

Reviewer #1 (Remarks to the Author):

This manuscript by Yang Li and others has been substantially improved, but it still requires amends.

Major points:

1. line 231 - how this new finding relates to "Ubiquitin Ligase Smurf1 Controls Osteoblast Activity and Bone Homeostasis by Targeting MEKK2 for Degradation" Cell, Volume 121, Issue 1, 8 April 2005, Pages 101–113? No references are included for Smad3/smurf1

I think that discussion is rather poor and should be amended. There is a very small number of references discussed and it needs to be improved

2. line 284 - what about "ubiquitin E3 ligase" term - was it used as well? Was the literature filtered based on some criteria, for instance were only peer-reviewed articles used for this search and was the data manually accessed in terms of quality of the evidence? Were only journals of good reputation used or also articles from 'predatory' journals? I checked the supplementary data and the papers I looked at were indeed of a very good quality, but I was not able to check all of them due to a high number of publications. Also, starting with line 753, there is a problem with column F, which does not display correct information anymore.

3. In my opinion, also reverse IP should be performed (IP: ubiquitin, IB: myc) . Figure 6D is not properly annotated and I am confused what it is trying to represent. Input lysate is not present there.

Minor points:

line 72 - 'pieces of evidence' - not 'evidences'

line 92 - literature

In general, there are still problems with English grammar - please have this manuscript edited professionally or by a native English speaker colleague.

3. Figure 6 - "(a-b)" should not be marked as bold. There are mistakes in the figure description such as immunoblotting (should be immunoblotting). Describe in the figure what CA means (catalytically dead mutant? if so, what residue was mutated).

4. Figure 6B - It is not clear what the gradient bar indicates above the figure.

Reviewer #2 (Remarks to the Author):

In this revised version, the authors have substantially improved the manuscript by performing additional experiments and providing a more detailed description of the methodology. Furthermore, additional experimental validation on a candidate prediction substantially strengthens the work. While it was hard for this reviewer to identify which parts of the manuscript had been changed, it appears that a majority of the earlier comments have been addressed satisfactorily. Interestingly it appears that due to these changes, the AUC has dropped from 0.90 to 0.84. This is likely a more realistic estimate of performance and is supported by other experiments (paired data cross-validation). However, there still are issues with the writing,

particularly, with respect to the use of informal language and grammatical errors. Detailed comments are provided below:

Minor:

- Although the authors have updated their cross-validation procedure to account for possible issues with paired data, the results suggest that when both the E3 ligase and the substrate are exclusively present in the training set, prediction performance drops substantially. The authors interpret this by comparing to the drops observed in the work of Park and Marcotte but clearly, these are not directly comparable. It is advised that statements related to this be modified.
- the use of terminology "naive Bayesian network" is nonstandard (or a credible source should be cited). Since individual lines of evidence are combined through multiplication, this corresponds to a naive Bayes classification, rather than a real Bayesian network. (the reviewer is aware that the naive Bayes can be seen as a simple Bayesian network, it is rather that the "naive Bayesian network" terminology is never used and can cause confusion here)
- Line 19: grammatical error – "it is an urgent need" should be "there is an urgent need."
- Line 347: typographical error – "negative" is misspelt.
- In general, it is recommended that a native speaker read the manuscript to check for errors.

Response to reviewer 1

Q1: This manuscript by Yang Li and others has been substantially improved, but it still requires amends. 1. line 231 - how this new finding relates to "Ubiquitin Ligase Smurf1 Controls Osteoblast Activity and Bone Homeostasis by Targeting MEKK2 for Degradation" Cell, Volume 121, Issue 1, 8 April 2005, Pages 101–113? No references are included for Smad3/smurf1. I think that discussion is rather poor and should be amended. There is a very small number of references discussed and it needs to be improved

A: Thanks for your great suggestion. Our finding is exactly consistent with Yamashita et al.'s study.

Yamashita et al found that the protein abundances of Smad(1,2,3,5) weren't influenced in *Smurf1*^{-/-} mice, but they also agreed that BMP pathway and TGF- β pathway were regulated in cells overexpressing Smurf1. (Yamashita et al , Cell, 2005, 121(1): 101-113.)

In our experiment, we found that Smad3 could interact with Smurf1, and Smurf1 mediated Smad3 ubiquitination in overexpression condition (Fig. 6c-e). Although Smad3 ubiquitination reduction was not obvious when *in vivo* Smurf1 was knocked out (Yamashita et al , Cell, 2005, 121(1): 101-113.), we considered that ubiquitination of Smad3 may be compensated by other ubiquitin ligases, such as Smurf2. Previous study has reported that Smad3 is a major substrate of Smurf2-mediated ubiquitination (Zhang Y et al., Nature 1996,383:168–172; Tang L et al, EMBO J. 2011,30: 4777–4789; Xu Z et al., Nat Commun. 2017 Feb 20;8:14570.). It has been suggested that Smurf2 may partially compensate the function for the loss of Smurf1. *Smurf1*^{-/-} and *Smurf2*^{-/-} (Smurf DKO) mice display embryonic lethality at around E12.5. However, single Smurf1 or Smurf2 mice have no overt defects in embryogenesis (Narimatsu M et al., Cell. 2009 Apr 17;137(2):295-307.). Therefore, the *in vivo* ubiquitination level change of Smad3 is difficult to be detected.

We also provided the related literature for the interactions between Smurf1 and Smad3. For example, Miriam Barrios-Rodiles's study shows the PPI between Smurf1 and Smad3 can be identified by high-throughput methods (Barrios-Rodiles M et al., Science, 2005, 307(5715): 1621-1625.). Takanori Ebisawa's study shows that there is weak interaction between Smurf1 and Smad3 (Ebisawa T et al., Journal of Biological Chemistry, 2001, 276(16): 12477-12480.).

Sorry for the poor description in the previous manuscript. We have added the above discussions and references in the revision.

Q2-1: line 284 - what about "ubiquitin E3 ligase" term - was it used as well?

A: Thanks for your suggestion.

In our study, "ubiquitin ligase" was used as the keyword to retrieve all the literature abstracts containing the information of potential E3 substrate interactions. We compared the papers retrieved using "ubiquitin E3 ligase" and those using "ubiquitin ligase". We found that 20180 papers were obtained using "ubiquitin E3 ligase" while 25190 using "ubiquitin ligase", and papers retrieved using "ubiquitin ligase" completely covered ALL the papers using "ubiquitin E3 ligases".

Therefore using "ubiquitin ligase" as keyword will get more complete sentences for subsequent manual curation and is appropriate in this paper.

Q2-2: Was the literature filtered based on some criteria, for instance were only peer-reviewed articles used for this search and was the data manually accessed in terms of quality of the evidence? Were only journals of good reputation used or also articles from 'predatory' journals? I checked the supplementary data and the papers I looked at were indeed of a very good quality, but I was not able to check all of them due to a high number of publications.

A: Thanks for your comments.

Yes, the literature was filtered under very strict criteria. We only used the peer

reviewed literature from PubMed database. And all these papers are published in journals of good reputation (The average impact factor of the related journals was 6.46).

We also checked whether there are papers from the predatory journals list. We found none of our source literature was included in the released predatory journal list (<https://web.archive.org/web/20161202192038/https://scholarlyoa.com/individual-journals/>). Thank you very much for your checking of our datasets, and it is really a hard work. In fact all records in the supplementary table were manually curated with double check, and we think that they are of high quality.

By the way, during this round of revision, we updated the golden standard positive dataset by strict manual curation. All the result of this manuscript kept unchanged against the updating.

Q2-3: Also, starting with line 753, there is a problem with column F, which does not display correct information anymore.

A: We are very sorry for the mistakes in the previous table.

This table has been corrected in the revision.

Q3: In my opinion, also reverse IP should be performed (IP: ubiquitin, IB: myc) .. Figure 6D is not properly annotated and I am confused what it is trying to represent. Input lysate is not present there.

A: Thanks for your great suggestion. Sorry for the poor description of Fig. 6d in the previous manuscript.

In Fig. 6d (revised version), we examined the interaction between Smurf1 and Smad3 by co-immunoprecipitation experiments. Smurf2 was used as a positive control and Smad3 was co-immunoprecipitated with Flag-Smurf1 and Flag-Smurf2. We have added the input lysate to this figure.

We also added the result of the reverse IP experiment for identifying the interaction between Smurf1 and Smad3 (Fig. 6e). The experimental result shows that the interaction between Smurf1 and Smad3 can be identified by both forward and reverse IP.

In addition, according to your suggestion we have corrected the annotation for Fig. 6d and added the lysate. Also, we have added Smurf2 as positive control.

For the *in vivo* Smad3 ubiquitylation assay, HA-ubiquitin, Flag-Smurf1 and Myc-Smad3, Smad1 were transfected, and ubiquitinated Smad3 was immunoprecipitated followed by immunoblotting with anti-HA. The ubiquitination experimental design follows the references of “Yamashita M et al., Cell. 2005 Apr 8;121(1):101-13.”, “Lu K et al., Nat Cell Biol. 2008 Aug;10(8):994-1002.”, “Tang L et al., The EMBO Journal (2011) 30, 4777–4789.” and “Xie P et al., Nat Commun. 2014 May 13;5:3733. doi: 10.1038/ncomms4733.”

Q4 : line 72 - 'pieces of evidence' - not 'evidences'line 92 – literature. In general, there are still problems with English grammar - please have this manuscript edited professionally or by a native English speaker colleague.

A : Thanks for your comments.

To improve the quality of the English writing, we have invited an English expert to help us to polish this manuscript.

Of course, we have corrected the relevant text errors.

Q5: Figure 6 - "(a-b)" should not be marked as bold. There are mistakes in the figure description such as immunoblotting (should be immunoblotting). Describe in the figure what CA means (catalytically dead mutant? if so, what residue was mutated).

A: We are very sorry for this mistake.

We have modified the figure legend of Fig. 6, and "a-b" is correctly marked in the

revision.

And the misspellings in the figure legend also has been corrected.

Yes, “CA” in Fig. 6b represents the “catalytically dead mutant”, and the mutated residue is Cysteine 699, which is a widely used Smurf1 inactivity mutant. Sorry for this misunderstanding abbreviation. We have changed “CA” to “C699A” in both Fig. 6b and manuscript.

Q6: Figure 6B - It is not clear what the gradient bar indicates above the figure.

A: We are very sorry for the mistakes during the figure labeling.

The gradient bar represents that the wild type Smurf1 is gradient transfected in the second and third band (not in the fourth band). The fourth band is transfected with C699A mutant Smurf1. Due to our mistakes, the gradient bar extended to the fourth band.

We have corrected this mistake.

Response to reviewer 2

Q1: In this revised version, the authors have substantially improved the manuscript by performing additional experiments and providing a more detailed description of the methodology. Furthermore, additional experimental validation on a candidate prediction substantially strengthens the work. While it was hard for this reviewer to identify which parts of the manuscript had been changed, it appears that a majority of the earlier comments have been addressed satisfactorily.

A: Many thanks for your comments for our revision on this manuscript.

For your convenience to identify which parts of the manuscript had been changed, we have provided the marked revised version as a separate file (“Main document marked with all changes.doc”).

Q2: Interestingly it appears that due to these changes, the AUC has dropped from 0.90 to 0.84. This is likely a more realistic estimate of performance and is supported by other experiments (paired data cross-validation).

A: Yes. We strongly agreed with you that the performance evaluation of our model is more realistic after the modification based on your suggestion.

Q3: Although the authors have updated their cross-validation procedure to account for possible issues with paired data, the results suggest that when both the E3 ligase and the substrate are exclusively present in the training set, prediction performance drops substantially. The authors interpret this by comparing to the drops observed in the work of Park and Marcotte but clearly, these are not directly comparable. It is advised that statements related to this be modified.

A: Thank you very much for your comments.

Yes, it is inappropriate to compare our results with Park and Marcotte’s result.

We have deleted the discussion about the comparison with Park and Marcotte’s result.

Q4: the use of terminology "naive Bayesian network" is nonstandard (or a credible source should be cited). Since individual lines of evidence are combined through multiplication, this corresponds to a naive Bayes classification, rather than a real Bayesian network. (the reviewer is aware that the naive Bayes can be seen as a simple Bayesian network, it is rather that the "naive Bayesian network" terminology is never used and can cause confusion here)

A: Thank you very much for your great suggestion.

We strongly agree with you that the term of “naïve Bayesian classification/classifier” should be used in our paper. And the term of “naïve Bayesian classification/classifier” has been widely used in literature, such as:

1. Friedman N, Geiger D, Goldszmidt M. Bayesian network classifiers[J]. Machine learning, 1997, 29(2-3): 131-163.
2. Flach P A, Lachiche N. Naive Bayesian classification of structured data[J]. Machine Learning, 2004, 57(3): 233-269.
3. Wang Q, Garrity G M, Tiedje J M, et al. Naive Bayesian classifier for rapid assignment of rRNA sequences into the new bacterial taxonomy[J]. Applied and environmental microbiology, 2007, 73(16): 5261-5267.
4. Rish I. An empirical study of the naive Bayes classifier[C]//IJCAI 2001 workshop on empirical methods in artificial intelligence. IBM New York, 2001, 3(22): 41-46.

We have corrected all the use of terminology "naive Bayesian network" in this revision.

Q5 : - Line 19: grammatical error – “it is an urgent need” should be “there is an urgent need.”

- Line 347: typographical error – “negative” is misspelt.

- In general, it is recommended that a native speaker read the manuscript to check for errors.

A: Thanks for your comments.

We are very sorry for these mistakes.

To polish this manuscript, we have invited an English expert to review our manuscript.

We hope the revised English can meet the requirement for publication.

REVIEWERS' COMMENTS:

Reviewer #1 (Remarks to the Author):

I think that the manuscript is now improved and ready for publication. I still have an issue with my previous question "Q6: Figure 6B - It is not clear what the gradient bar indicates above the figure."

"A: We are very sorry for the mistakes during the figure labeling.

The gradient bar represents that the wild type Smurf1 is gradient transfected in the second and third band (not in the fourth band)."

Since there is a gradient band, does it imply that wt Smurf is in more concentration in the third band and less in the second band? If not, I would not use a gradient bar but a solid line.

Thank you!

Response to reviewer 1

Q: I think that the manuscript is now improved and ready for publication. I still have an issue with my previous question "Q6: Figure 6B - It is not clear what the gradient bar indicates above the figure."

A: We are very sorry for the mistakes during the figure labeling.

The gradient bar represents that the wild type Smurf1 is gradient transfected in the second and third band (not in the fourth band)."

Since there is a gradient band, does it imply that wt Smurf is in more concentration in the third band and less in the second band? If not, I would not use a gradient bar but a solid line.

A: Just as your kind comments, it is indeed that wt Smurf is in more concentration in the third band and less in the second band because these two bands are with different Flag-Smurf1-WT transfection. During the transfection, to detect whether Smurf1 reduced the stability of Smad3 in a dose-dependent manner, we transfected 0.5 μ g of Flag-Smurf1-WT into the second and 1 μ g into the third one (the fourth one of Flag-Smurf1-CA was also 1 μ g).

Thanks for your careful checks and reminding. We've revised the figure legend for Figure 6B to avoid the confusion.